# Extensive horizontal gene transfer in cheese-associated bacteria

Kevin S Bonham[1], Benjamin E Wolfe[2], Rachel J Dutton[1,3]*

[1]Division of Biological Sciences, University of California, San Diego, San Diego, United States; [2]Department of Biology, Tufts University, Medford, United States; [3]Center for Microbiome Innovation, Jacobs School of Engineering, University of California, San Diego, San Diego, United States

**Abstract** Acquisition of genes through horizontal gene transfer (HGT) allows microbes to rapidly gain new capabilities and adapt to new or changing environments. Identifying widespread HGT regions within multispecies microbiomes can pinpoint the molecular mechanisms that play key roles in microbiome assembly. We sought to identify horizontally transferred genes within a model microbiome, the cheese rind. Comparing 31 newly sequenced and 134 previously sequenced bacterial isolates from cheese rinds, we identified over 200 putative horizontally transferred genomic regions containing 4733 protein coding genes. The largest of these regions are enriched for genes involved in siderophore acquisition, and are widely distributed in cheese rinds in both Europe and the US. These results suggest that HGT is prevalent in cheese rind microbiomes, and that identification of genes that are frequently transferred in a particular environment may provide insight into the selective forces shaping microbial communities.

## Introduction

Great strides have been made in characterizing the composition of microbiomes, and in understanding their importance in the ecology of natural systems, in agriculture and in human health. However, despite these advances, the forces that shape the diversity, structure, and function of microbiomes remain poorly understood (*Widder et al., 2016*). Investigating these underlying mechanisms in situ is difficult, as observational and sequenced-based analysis rarely enables causal conclusions (*Nemergut et al., 2013*). Replicating microbial communities in vitro is also an enormous challenge, because of high levels of diversity and the difficulties in establishing pure cultures of most bacterial species. These obstacles significantly hamper our ability to move from observations of microbial diversity to the molecular mechanisms shaping key processes such as species interactions and microbial evolution.

Horizontal gene transfer (HGT) is a major force in microbial evolution and can lead to the wholesale acquisition of novel functions. In some cases, these novel functions can have significant adaptive consequences, such as in the transfer of antibiotic resistance genes (*Ochman et al., 2000*). HGT also allows rapid adaptation to new niches (*Wiedenbeck and Cohan, 2011*), as ecologically relevant genes may be acquired by species not previously adapted to a particular niche (*Tasse et al., 2010*; *Hehemann et al., 2010*). The movement of microbes to new environments has been shown to increase both the rate and impact of HGT, and HGT is most frequent for genes under positive selection (*Niehus et al., 2015*). In moving to a new environment, microbes can face novel abiotic conditions (temperature, moisture, salinity, pH, and nutrients) and novel biotic challenges and opportunities resulting from the presence of microbial neighbors.

Evaluating HGT within the context of microbial communities could provide new insights concerning the extent, mechanisms, and ecological impact of this important process. Advances in genome

*For correspondence: rjdutton@ucsd.edu

Competing interests: The authors declare that no competing interests exist.

**eLife digest** From the depths of the ocean to the lining of the human gut, almost every environment on Earth is home to a unique community of microorganisms referred to as a microbiome. Within these communities, unrelated microorganisms can exchange genetic information through a process known as horizontal gene transfer. For example, genes linked to antibiotic resistance are often transferred between different microorganisms, which can create increasingly drug resistant microbes and has important implications for human health.

Horizontal gene transfer has been studied for almost 100 years, but examining it directly is challenging because, almost by definition, it requires studying a community of microbes rather than one microbe in isolation. As such, researchers are looking for simple models of microbial communities that can be easily manipulated in experiments.

Bonham et al. have now turned to the outer surface of cheese, also known as cheese rind, to better understand horizontal gene transfer. As a model system, the cheese rind microbiome is relatively simple to work with because cheese rind is easy to replicate in the laboratory, and the microbes growing on cheese can be grown on their own or in combinations with other microbes. By comparing the genetic material of 165 cheese-associated bacteria to one another, Bonham et al. identified over 4,000 genes that were shared between the bacteria, including several large clusters of genes that were shared by many species.

Many of the identified genes (about 23% to be precise) appear to help the microorganisms acquire nutrients that are known to be in short supply on the surface of cheese surface, including iron. Bacteria typically use specialized molecules called siderophores to scavenge for iron and uptake systems to carry the iron-bound siderophore back into the cell. Notably, only the genes associated with the uptake systems were found in some of the shared gene clusters. This finding suggests that horizontal gene transfer has allowed some microbes to "cheat" and take up iron-bound siderophores without expending energy to produce the siderophores themselves.

Using the cheese rind microbiome as a model system, it becomes possible to explore how horizontal gene transfer works in more detail than before. A better understanding of this process can then be applied to other important microbiomes, including those where genes conferring antibiotic resistance are commonly exchanged.

sequencing have begun to provide a glimpse into HGT within environmentally, medically and economically important microbiomes (*McDaniel et al., 2010*; *Andam et al., 2015*). For example, extensive gene sharing has been observed throughout the commensal human microbiome, including sharing of genes that enable nutrient acquisition from novel food sources (*Hehemann et al., 2010*; *Smillie et al., 2011*), and pathogenicity islands and antibiotic resistance genes in pathogenic microbes (*McCarthy et al., 2014*; *Hiramatsu et al., 2001*; *Forsberg et al., 2012*). There is evidence of extensive HGT in other natural habitats, such as soil (*Coombs and Barkay, 2004*; *Heuer and Smalla, 2012*) and aquatic environments (*McDaniel et al., 2010*; *Frischer et al., 1994*). Although these studies offer valuable insights into the frequency and potential impact of genes that can be transferred in microbial communities, the complexity of these systems makes difficult any further examination of the effects of these HGT events on their evolution and ecology.

The microbial communities of fermented foods experience strong selection as a result of growing in novel, human-made environments. Previous work has demonstrated that HGT can be a major driver of adaptation in food systems and other human-managed environments (*Andam et al., 2015*; *Rossi et al., 2014*). Prior analysis of microbial species from cheese has revealed several instances of HGT in this environment. Lactic acid bacteria (LAB) such as *Lactobacillus* and *Lactococcus*, which are used in the initial fermentation of milk, are known to harbor antibiotic resistance genes and may be reservoirs for transfer to pathogenic enterococci (*Wang et al., 2006*; *Mathur and Singh, 2005*) and other pathogenic microbes. Other food-associated bacteria may also contribute to antibiotic resistance gene transfer (*Cocconcelli et al., 2003*; *Delorme, 2008*; *Flórez et al., 2005*). In yogurt, another dairy ferment using LAB, HGT of metabolic genes has been observed between

protocooperative species *L. bulgaricus* and *S. thermophilus* (*Li et al., 2013*; *Liu et al., 2009*). Sequencing of *Penicillium* species isolated from the cheesemaking environment identified HGT of large genomic islands between these key fungal inhabitants of cheese (*Cheeseman et al., 2014*; *Ropars et al., 2015*; *Gibbons and Rinker, 2015*).

During the aging of traditional styles of cheese in caves or aging rooms, bacteria and fungi form a multi-species biofilm called the rind (*Button and Dutton, 2012*). We have previously shown that these communities can be used to examine community-based processes, such as succession and interspecies interactions, within an experimentally tractable system (*Wolfe et al., 2014*; *Kastman et al., 2016*). Given that biofilms such as these are densely populated, and microbes in cheese rinds are under strong selection to obtain limited nutrients (e.g. free amino acids, iron) as well as tolerate cell stress (*Monnet et al., 2015*), we predicted that HGT might be widespread in cheese rind microbiomes and therefore might provide a useful experimental model for HGT within microbial communities.

We sought to determine the diversity, distribution, and functional content of HGT in bacterial species isolated from cheese rinds. Specifically, we predicted that (1) HGT would be widespread, (2) HGT genes would be enriched for functions related to survival in cheese environment, and (3) there would be uneven distribution of HGT events across taxa. We analyzed the genomes of newly isolated and sequenced cheese-associated bacterial species (31 genomes) and those available in public databases (134 additional genomes). We present data which suggest that there is extensive HGT in cheese-associated bacteria. The regions of DNA identified appear to encode a number of functions which would be expected to provide adaptive advantages within the cheese environment. In particular, we identified three large multi-gene islands that are shared within multiple Actinobacteria, Proteobacteria, and Firmicutes species. These genomic regions are not related, but appear to have analogous functions involving iron acquisition, and are widely distributed in geographically distant cheeses. This work provides foundational knowledge in an experimentally tractable system in which future work could help to provide insight into the role of HGT within microbiomes.

## Results

### Identification of putative horizontally transferred regions

To establish a diverse database of cheese-associated bacterial genomes, we isolated species from cheese samples collected as part of previous work (*Wolfe et al., 2014*). A total of 31 isolates, representing four bacterial phyla and 11 genera, were selected for genome sequencing using Illumina and PacBio (*Supplementary file 1a*). Recently, a large collection of cheese-associated bacterial genomes were sequenced (*Almeida et al., 2014*), allowing inclusion of additional genomes in our analysis. Our isolates were from cheeses produced in the United States, Spain, Italy, and France, while the Almeida et al. collection was almost exclusively from France. We also included genomes from the NCBI reference sequence (RefSeq) database that are associated with cheese, for a total of 165 bacterial genomes.

We next developed a computational pipeline for identification of putative horizontally transferred genes, adapted from work on the human microbiome (*Smillie et al., 2011*). We built a central BLAST database containing all ORFs from all cheese-associated genomes. For each gene in each genome, we performed BLAST against this database, and compiled a list of hits (*Figure 1—figure supplement 1*, Materials and methods). For each hit, we examined the length and percent identity of aligned regions. Closely related species will have many nearly identical genes as a result of vertical inheritance. To avoid capturing these genes in our analysis, we determined the pairwise average nucleotide identity (ANI) between species within the same genus (*Chan et al., 2012*; *Rodriguez-R and Konstantinidis, 2016*). ANI provides a measure of the overall similarity of two genomes. We tested varying thresholds for length and ANI to examine the effects of these parameters on the results (*Supplementary file 1c*). Higher maximum ANI cutoffs and shorter lengths are more likely to yield false positives, as closely related species are more likely to share short stretches of high nucleotide identity. At the same time, a lower maximum ANI cutoff may exclude legitimate HGT events, especially considering that closely related species are also more likely to engage in HGT. Based on our most conservative gene identity parameters (minimum 99% identity over 500 nucleotides), we identified at least one putative horizontally transferred gene in 130/165

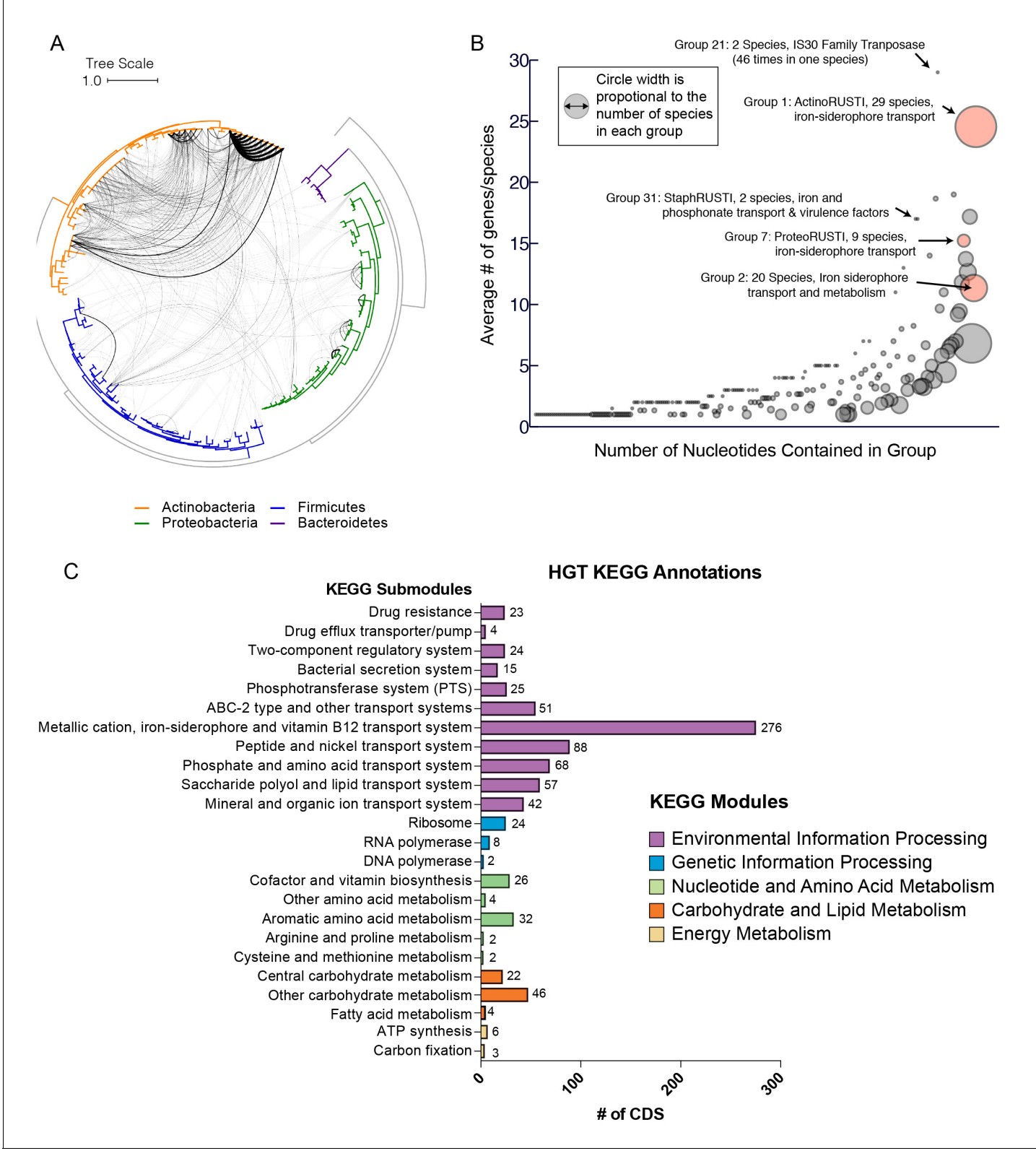

**Figure 1.** Extensive horizontal gene transfer in the cheese microbiome. (**A**) All HGT events in analyzed cheese-associated bacteria. Connection thickness is scaled to number of shared protein coding sequences. Maximum likelihood tree based on 16S RNA alignment using Ribosomal Database Project (RDP). (**B**) HGT events clustered into 264 'groups' based on genomic proximity. Groups are plotted based on total nucleotide content (x-axis, from low to high), and the mean number of genes per species (y-axis). Diameter of each circle is proportional to the total number of species in the

*Figure 1 continued*

group. Groups highlighted in red are described further in the text. (**C**) Quantification of KEGG modules and submodules for protein coding genes (CDS) identified as horizontally transferred. Annotations were generated by BLAST-Koala. Genes without function prediction are not depicted.

The following figure supplements are available for figure 1:

**Figure supplement 1.** Schematic of software pipeline to identify HGT.

**Figure supplement 2.** Same as *Figure 1A* with branch labels.

(78.8%) cheese-associated species in the analysis, for a total of 4733 genes (*Figure 1A*, *Supplementary file 1d*). Because this analysis depends on the species included for comparison, this list of HGT is almost certainly an underestimate.

As multiple genes can be transferred in a single HGT event, we next assembled the putative HGT genes into groups based on genomic proximity. Individual coding sequences (CDS) for each species were grouped into islands if they were found within 5000 nucleotides of one another on the same contig. These islands were then clustered with islands in other species that shared at least one CDS. The 4733 genes were clustered into 264 individual groups (*Figure 1B*, *Supplementary file 1d*). Mobile elements such as transposons complicate our method of group clustering, as non-contiguous islands may be grouped together if they share a common transposon. Indeed, this appears to have occurred with Group 1, which contains genes from several disparate genomic regions. In other cases, a single species may have genes in a single group spread across multiple contigs (*Figure 2—figure supplement 1*), but this may accurately represent a single HGT event.

Most HGT groups identified (231, 87.5%) contain only members of the same phylum, or even a single genus (183 or 69%, *Supplementary file 1d*). This supports previous studies which suggest that HGT is most prevalent among closely related species (*Ravenhall et al., 2015*). However, we uncovered several notable exceptions. For example, *Alkaibacterium kapii* FAM208.38, a Firmicute, has a substantial (~8 kb) fraction of Group 1, which is predominantly found in *Actinobacteria* species. Groups 2 and 3 each have hits found in three different phyla (although both are predominantly found in Actinobacteria).

## Functions encoded in HGT regions

HGT can enable rapid evolution of microbes entering a new environment, with genes that are under positive selection observed more frequently (*Wiedenbeck and Cohan, 2011*). Identifying the functions of genes that are frequently transferred could help to identify the selective forces that are most important for adapting to the cheese rind environment. Across all of the genes identified in our analysis, the most abundant gene functions are transposases, conjugal transfer genes, phage-related proteins, and other mobile elements (631/4733 or 13% of all protein coding sequences). A third (86/264 or 32.6%) of all HGT groups contain mobile elements. These genetic elements are likely involved either directly or indirectly in mobilization and transfer of DNA.

To determine whether gene functions other than mobile elements are enriched in identified HGT regions, we used BlastKOALA (*Kanehisa et al., 2016*) to assign KEGG functional annotations (*Figure 1C*, *Figure 2—figure supplement 1c*). In approximately half (53%) of the genes, KEGG annotations could not be assigned. Of the KEGG-annotated genes, the most frequent module (281/2264 or 11%) was 'metal ion, iron siderophore and vitamin B12 transport systems'. Five of the 10 largest HGT groups as measured by total number of genes (Groups 1, 2, 3, 7 and 8) contained siderophore transport systems (K02013-K02016). Low availability of iron in cheese is known to limit the growth of several bacterial species (*Monnet et al., 2012*, *2010*). Previous work has also shown that genes involved in iron acquisition are present in higher numbers in cheese-associated species compared with closely related species from other environments (*Monnet et al., 2010*; *Walter et al., 2014*).

Many other horizontally transferred genes (267/2264 or 12% of KEGG-annotated genes) are also involved in transport of nutrients relevant for growth in the cheese environment. Lactate is an abundant carbon source in freshly made cheese, as the initial stages of cheesemaking involve

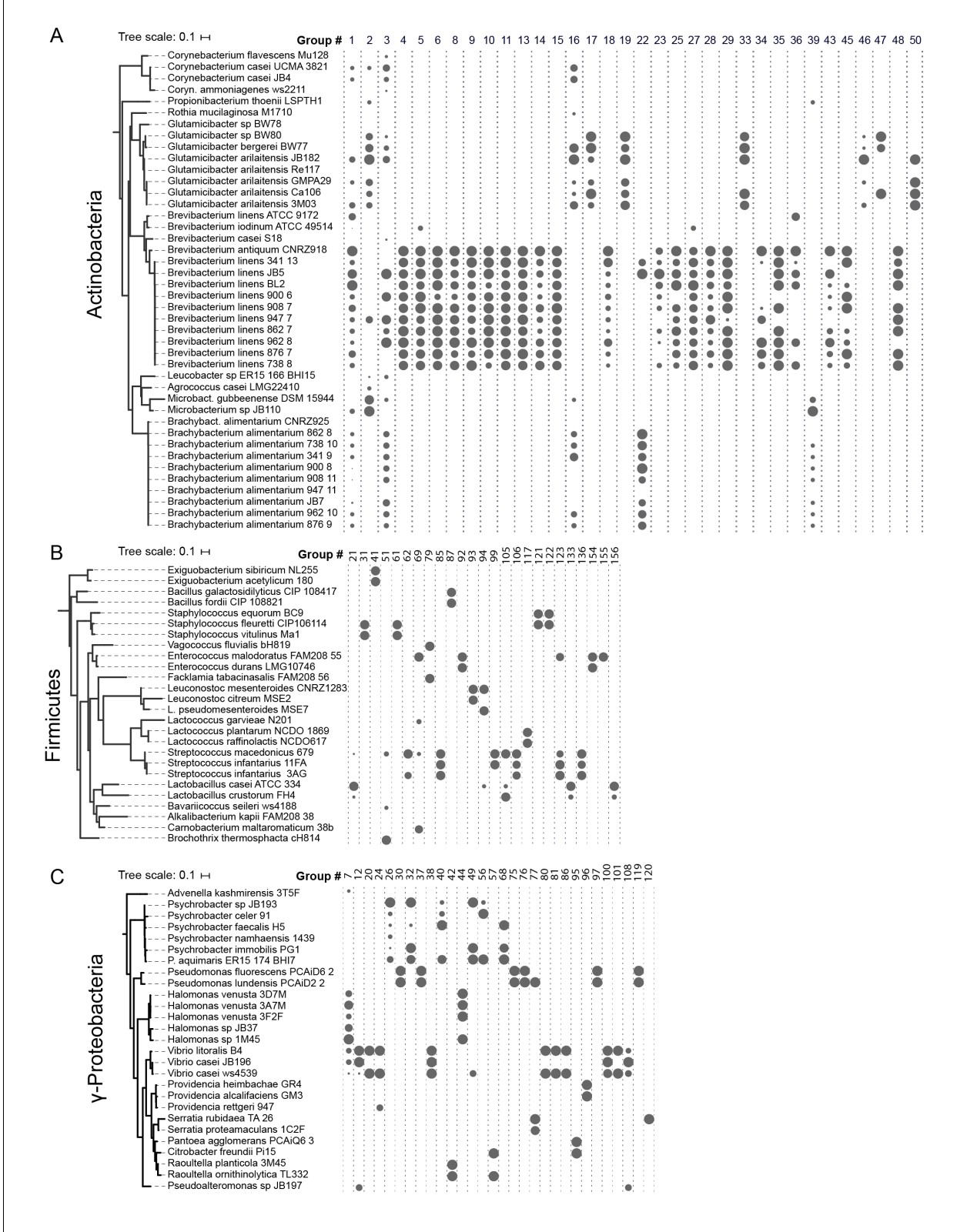

**Figure 2.** HGT Groups in Actinobacteria, Firmicutes, and γ-Proteobacteria groups. (**A**) The 31 largest HGT groups that contain predominantly Actinobacteria. The areas of circles are scaled to log2(n), where n is the total number of nucleotides in that group for each species. The largest circle size represents the largest HGT group in that phylum. Phylogenies (left) are based on small subunit ribosomal RNA alignment. (**B**) The 25 largest HGT groups that contain predominantly Firmicutes. (**C**) The 28 largest groups that contain predominantly γ-Proteobacteria.

*Figure 2 continued on next page*

*Figure 2 continued*

The following figure supplement is available for figure 2:

**Figure supplement 1.** Group A: Expected clustering: contiguous genes in multiple species are in a single group.

fermentation of lactose to lactate by lactic acid bacteria (*Button and Dutton, 2012*). We observed a large number of genes (63/2264 or 2.8% of KEGG-annotated genes) involved in lactate import (lactate permease - K03303) or lactate metabolism. Lactate dehydrogenase (K00101), which reduces lactate to pyruvate, represented nearly 1% of all horizontally transferred protein coding sequences.

Apart from lactate, the primary source of energy for microbial growth in cheese is metabolism of the abundant lipids and proteins, particularly casein (*Monnet et al., 2015*). We also identified glutamate importers (43/2264 or 1.9%, eg. K12942, K10005-K10008) and short peptide/nickel transporters (88/2264 or 3.9% eg. K03305) in our analysis, suggesting pathways for utilization of casein degradation products. Transporters for micronutrients, including phosphonate (K05781, K06163-K06165), molybdate (K02017, K02019, K02020, K03750, K03750, K03639), and metal ions like zinc and manganese were also identified.

HGT of drug resistance genes is of particular concern, as mobile resistance genes from food-associated microbes may be transferred to animal- and human-associated microbes (*Rossi et al., 2014*). Cheese rind communities often contain filamentous fungi including *Penicillium* species and other microbes that could potentially produce antimicrobial compounds and thus select for antibiotic resistance in co-occurring species. However, less than 1% of KEGG-annotated genes in this dataset are related to drug resistance. A tetracycline resistance gene was identified in eight *Brevibacterium* species (group 10) and a tripartite multidrug resistance system (K03446, K03543) in three *Pseudomonas* species (group 37).

We also noted a small number of genes from the core genome that were not expected to be horizontally transferred. For example, group 27 is found in all 10 strains of *B. linens* in this dataset, as well as the closely related *B. antiquum* CNRZ918, and contains the SSU ribosomal protein S1p, as well as DNA Polymerase 1. It is possible that these results are false positives as *B. linens* and *B. antiquum* have an ANI ~88%, and these genes are typically more highly conserved than average. At the same time, other ribosomal genes that should also be highly conserved protein coding genes have substantially lower homology between these species than S1p (*Supplementary file 1e*). Further, another gene within this HGT group (SAM-dependent methyltransferase) is not typically highly conserved, but nevertheless is >99% identical between these *Brevibacterium* species. We cannot exclude the possibility that this is a false positive, but this may be an example of homologous recombination facilitated by the high sequence identity of the ribosomal protein gene. Several other groups also contain ribosomal proteins (42 L5p and S3p, 180 L4p, 219 S3p), but these groups do not contain any other protein coding genes, and they are clustered with other ribosomal protein coding genes, which is a more typical arrangement.

## Iron acquisition HGT

The abundance of iron acquisition genes identified as HGT suggests that iron is a driving force in adaptation to growth on cheese. The largest HGT region we identified in cheese-associated bacteria, Group 1, includes an island of ~47 kbp (~1% of the genome of *B. linens* JB5) and 34 genes. This island is found in whole or in part in 15 different species in five different Actinobacterial genera (*Brachybacterium, Brevibacterium, Corynebacterium, Microbacterium,* and *Glutamicibacter,* formerly *Arthrobacter*), and one Firmicute (*Alkalibacterium*). The core of this region, flanked by AraC-like transcriptional regulators (e.g. Ga0099663_102740 and Ga0099663_102753 from JB182), contains several genes predicted to form a siderophore import complex, including two cell-surface associated substrate binding protein genes (Ga0099663_102743–44), two membrane permease genes (Ga0099663_102745–46), and an ATPase subunit (Ga0099663_102747). A siderophore reductase (Ga0099663_102741) is present immediately downstream of the AraC regulator, but has less than 99% identity between the species we analyzed (*Figure 3A*, *Supplementary file 1c*). We named this region i**R**on **U**ptake/**S**iderophore **T**ransport **I**sland (RUSTI).

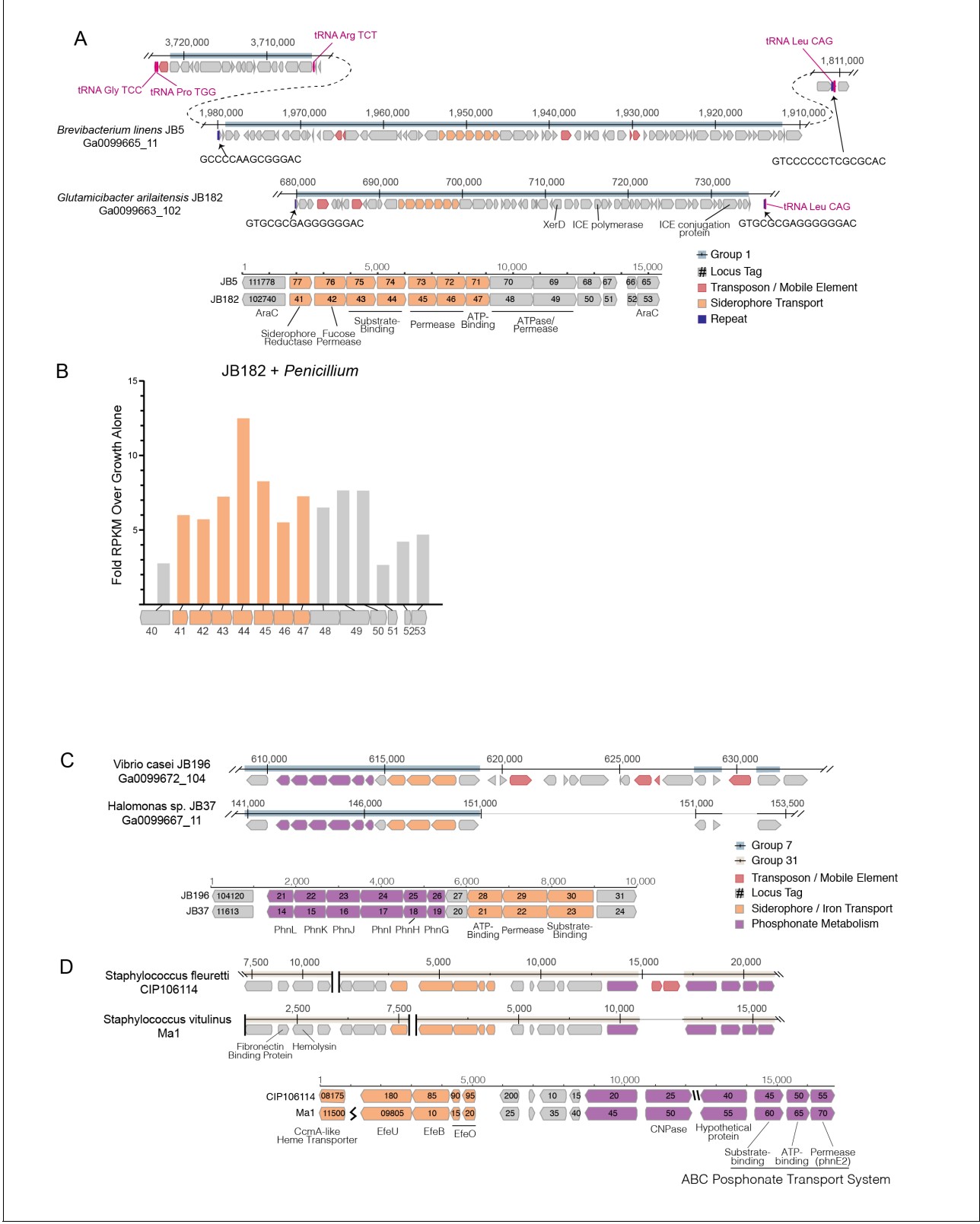

**Figure 3.** Structure of RUSTI islands. (**A**) At-scale schematics for genomic context of HGT Group 1 (top) for *B. linens* JB5 and *G. arilaitensis* JB182 and alignment of RUSTI operon (bottom). Nucleotide position values (top) refer to contigs Ga0099665_11 and Ga0099663_102 respectively. Dotted line for JB5 depicts regions of the contig that are not shown. Nucleotide position values (bottom) refer to operon starting from stop codon of leading AraC coding sequence. (**B**) At-scale schematics for genomic context of HGT Group 7 for *Halomonas sp.* JB37 and *V. casei* JB196 (top) and alignment of iron

*Figure 3 continued on next page*

*Figure 3 continued*

and phosphonate metabolism genes (bottom). Nucleotide position values (top) refer to contigs Ga0099667_11 and Ga0099672_104 respectively. Grey lines for JB196 depict gaps in the alignment resulting from insertions in JB37. Nucleotide position values (bottom) refer to operon starting from stop codon of leading protein coding sequence. (C) At-scale schematics for genomic context of HGT Group 31 for *S. fleuretti.* CIP106114 and *S. vitulinus* Ma1 (top). For both species, the group is split across two different contigs and nucleotide position values (top) refer to the relative position for that contig. Alignment of iron and phosphonate metabolism genes from Group 31 (bottom).

Horizontally transferred genes are not always expressed in the recipient genome, because of possible incompatibilities in promoter sequence (*Ochman et al., 2000*). As iron is a limiting resource in cheese (*Kastman et al., 2016*; *Monnet et al., 2012*), we reasoned that if RUSTI is a functional operon, it would likely have increased expression in the presence of additional competition for iron. To assess whether RUSTI genes are regulated in the presence of competition, we grew *G. arilaitensis* JB182 alone or in the presence of *Penicillium* and performed RNA sequencing (RNA-seq) to monitor gene expression. The genes in RUSTI were significantly upregulated in the presence of a competing microbe relative to growth alone (*Figure 3B*, *Supplementary file 1f*), suggesting that this horizontally transferred region is transcriptionally active and may be responding to competition for limited iron in cheese.

Hundreds of different siderophores have been identified belonging to three major classes: hydroxamate, catecholate, and α-hydroxycarboxylate (*Hider and Kong, 2010*). To predict the function of the RUSTI transporters, we compared their protein sequences to the Transporter Classification Database (TCDB) (*Saier et al., 2016*) using BLAST (*Supplementary file 1g*). The two genes annotated as permease subunits and one gene annotated as an ATP binding subunit each share substantial homology with the catechol ferric enterobactin transport system (FepD, FepG, and FepC, respectively) in *E. coli* (*Elkins and Earhart, 1989*; *Shea and McIntosh, 1991*; *Chenault and Earhart, 1991*). Two genes annotated as substrate binding proteins have weak homology to vibriobactin and iron(III) dicitrate binding proteins from *Vibrio cholerae* and *E. coli*, respectively.

Siderophore-related genes are also well-represented in γ-Proteobacterial HGT groups. Like Group 32 in Actinobacteria, Group 39 contains both siderophore acquisition and siderophore biosynthesis genes and is found in three species of *Psychrobacter*. The HGT group with the most protein coding genes that we identified in γ-Proteobacteria (Group 7) is found in several *Vibrio* and *Halomonas* species, and like ActinoRUSTI contains an ABC siderophore transport system with individual substrate-binding, permease, and ATP-binding domains (*Figure 3C*). Although this group appears to have analogous function in the acquisition of iron with RUSTI from Actinobacteria, this ProteoRUSTI does not appear to be related. TCDB analysis suggests homology to hemin transporters in *Yersinia pestis* and *Bordetella pertussis* (*Supplementary file 1g*).

The same gene island also contains genes related to the Phn family involved in phosphonate import and metabolism (*Jochimsen et al., 2011*). Phosphonate metabolism genes have previously been associated with iron siderophore acquisition in acidic environments (*Osorio et al., 2008*), and cheese is typically close to pH5 during the initial periods of rind community growth. Interestingly, BLAST of this region against the NCBI RefSeq database reveals that several uropathogenic *E. coli* strains share identical DNA sequences (*Supplementary file 1h*). Highly similar sequences are found in *Oligella urethralis*, another gram-negative pathogen of the urogenital tract, and *Vibrio harveyi*, a bioluminescent ocean-dwelling microbe. Iron sequestration by animals is a common defense against pathogens (*Parrow et al., 2013*), and enhanced iron acquisition is commonly associated with virulence. Mammals produce lactoferrin in milk for the same reason (*Ellison, 1994*), and these data suggest that the same genes would be adaptive in both pathogenesis and growth on cheese.

The convergence of strategies for both pathogenesis and growth on cheese was also demonstrated in the Firmicutes (*Figure 3D*). Two species of cheese-associated *Staphylococcus* (*S. fleuretti* CIP106114 and *S. vitulinus* Ma1) share a large (~20 kb) cluster of genes (Group 31) that includes hemolysin and fibronectin binding protein (FnBP), which are involved in virulence in *S. aureus* (*Sinha et al., 1999*; *Cheung and Ying, 1994*). Hemolysin (also known as alpha toxin) forms pores in cell membranes, and is so-called because of its ability to lyse red blood cells. FnBP enables binding to and invasion of cells, and has been implicated in formation of biofilms in methicillin resistant *S. aureus* (*McCourt et al., 2014*). It is unlikely that these genes provide a selective advantage to

cheese-associated Staphylococci, but Group 31 also contains genes for iron acquisition. These genes are homologous to the EfeUOB systems in *E. coli* and *Bacillus subtilis* and FepABC system in *S. aureus,* which are active in low-pH conditions (*Cao et al., 2007*; *Miethke et al., 2013*; *Turlin et al., 2013*). These iron acquisition genes are also found in association with hemolysin and FnBP in *S. aureus* and the animal-associated *S. sciuri*, although at only ~80% nucleotide identity (*Supplementary file 1h*) (*Kloos and Schleifer, 1976*). As iron acquisition genes may be adaptive both on cheese and in pathogenesis, it is possible that this region was acquired from an animal path-ogen, and the virulence genes have been preserved because of their association with these genes.

The genomes sequenced are from a limited number of cheeses from Europe and the United States. To determine the distribution of RUSTI across a more expansive sample of cheese micro-biomes, we used BLAST searches against assembled metagenomic data from 38 different cheeses using representative Proteo-, Actino-, and Staph-RUSTI sequences (*Figure 4*). Gene islands at least 97% identical to ActinoRUSTI were readily identified in 23 (61%) of these metagenomes, in both nat-ural and washed rind cheeses from the United States and Europe. Although less common (26% of metagenomes), ProteoRUSTI was also identified in diverse cheeses in both the US and Europe. StaphRUSTI could not be found in any of the metagenomes we analyzed. These data demonstrate that siderophore-associated HGT islands are widespread in cheese rind microbiomes. Whether inde-pendent HGT events occur within each cheese production and aging facility, or whether they occurred before the widespread distribution of these microbes across cheese produc-tion regions is unknown.

## A potential mechanism of transfer and source of the Actinobacterial RUSTI

To begin to understand potential mechanisms which could mediate HGT in cheese-associated bacte-ria, we analyzed the sequences surrounding the RUSTI region of *Glutamicibacter* JB182. Conjugative elements are commonly involved in HGT (*Wozniak and Waldor, 2010*). Integrative and conjugative elements (ICEs) can in part be identified by the presence of signature proteins associated with core functions of integration into and excision from the host genome (recombinase), replication as an extrachromosomal element (polymerase), and conjugation from the host to recipient cell (conjuga-tion) (*Ghinet et al., 2011*). Analysis of the *Glutamicibacter* JB182 RUSTI region revealed homologs of each of these protein classes (*Figure 3A*): a recombinase of the site-specific tyrosine recombinase XerD family (Ga0099663_102762) (*Subramanya et al., 1997*), a hexameric ATPase conjugation pro-tein of the VirD4/TraG/TraD family (Ga0099663_102784) (*Hamilton et al., 2000*), and a homolog of the bi-functional primase-polymerase DNA replication protein family (Ga0099663_102766). Interest-ingly, Actinobacterial ICE systems typically use conjugation apparatus belonging to the SpoIIE/FtsK family, which allows transfer of double-stranded DNA (*te Poele et al., 2008*; *Bordeleau et al., 2012*). However, the conjugation machinery here is more reminiscent of gram-negative and Firmi-cute systems of single-stranded transfer (*Burrus and Waldor, 2004*).

ICE integration is site-specific, and frequently occurs at the 3' end of tRNA genes (*Ghinet et al., 2011*). Immediately downstream of the RUSTI region in *Glutamicibacter* is a leucine tRNA. The 3' end of the tRNA forms an imperfect repeat with the region immediately upstream of the RUSTI region, which strongly suggests that the tRNA-Leu is used at the integration site (*att* site) for this ICE. To determine whether this ICE was still active, we performed PCR using primers within and flanking the putative integration site (*Figure 5A*). We were able to detect PCR products which sug-gest that at least a portion of the cells within the population have lost the RUSTI ICE from their chro-mosome, and it is present as an extrachromosomal circular form (*Figure 5B*). Sequencing of the PCR product (primers 1 + 6) that spans the predicted excision site matched the predicted remaining sequence, containing Repeat element B (*Figure 5D*). Sequencing of the PCR product (primers 2 + 5) that spans the predicted circularization site matched the predicted sequence, containing Repeat ele-ment A (*Figure 5E*).

There are several possible explanations for the widespread distribution of nearly identical Actino-RUSTI. Initial transfer events may have occurred in a single location, on the surface of a cheese or in livestock, and subsequently been dispersed to many separate cheesemaking facilities. The continued mobility of the ICE in JB182 raises the alternate possibility that it may be continually introduced to many cheeses from a common source. Cheese producers often use commercially available 'starter' cultures that contain desirable species, including many Actinobacteria (*Robinson, 2005*). We tested

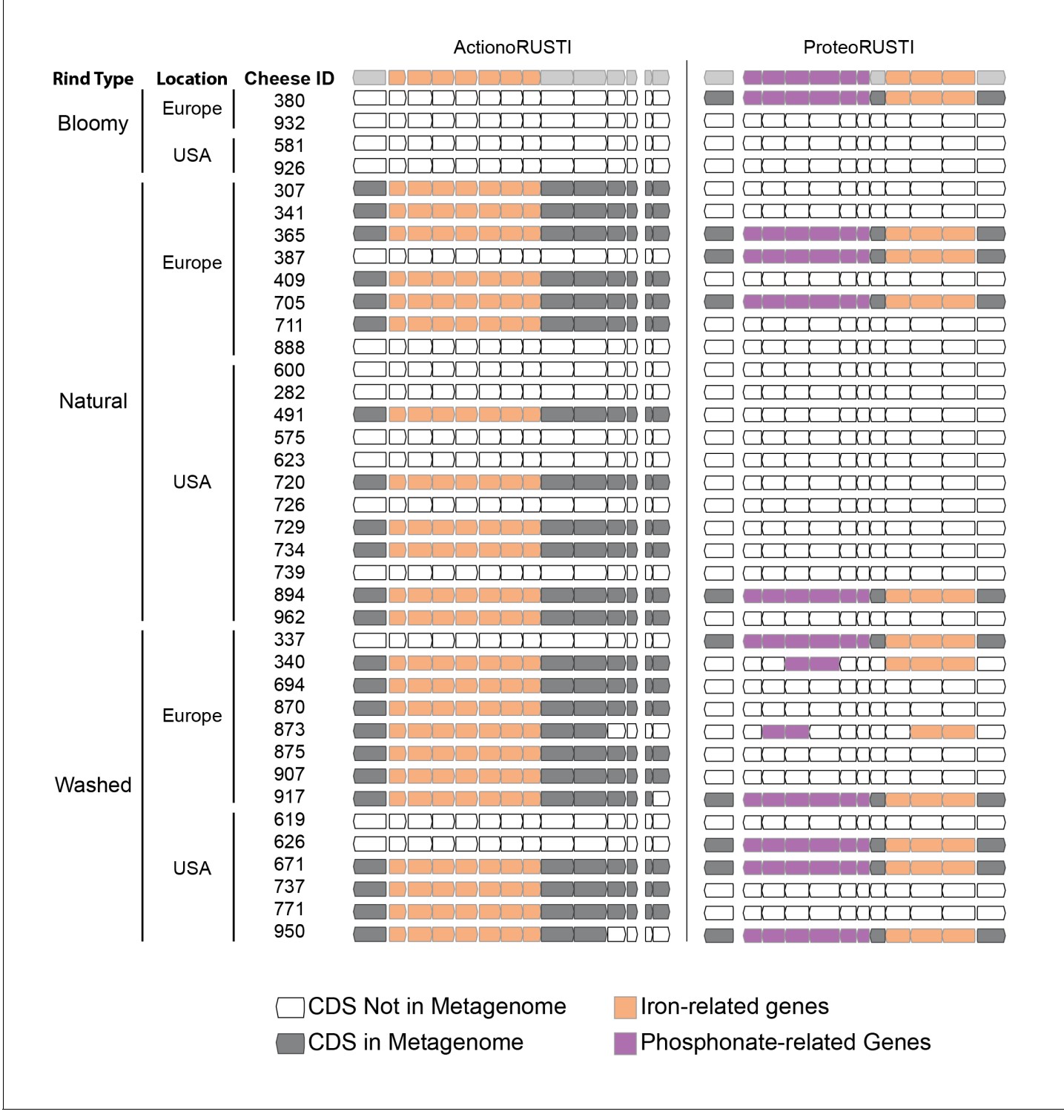

**Figure 4.** Presence of RUSTI in cheese metagenomes. Genes in ActinoRUSTI (*G. arilaitensis* JB182) and ProteoRUSTI (*V. casei* JB196) regions were compared with 32 assembled metagenomes from the US and Europe. Filled CDS represents a positive (>97% identical nucleotides) hit in that metagenome.

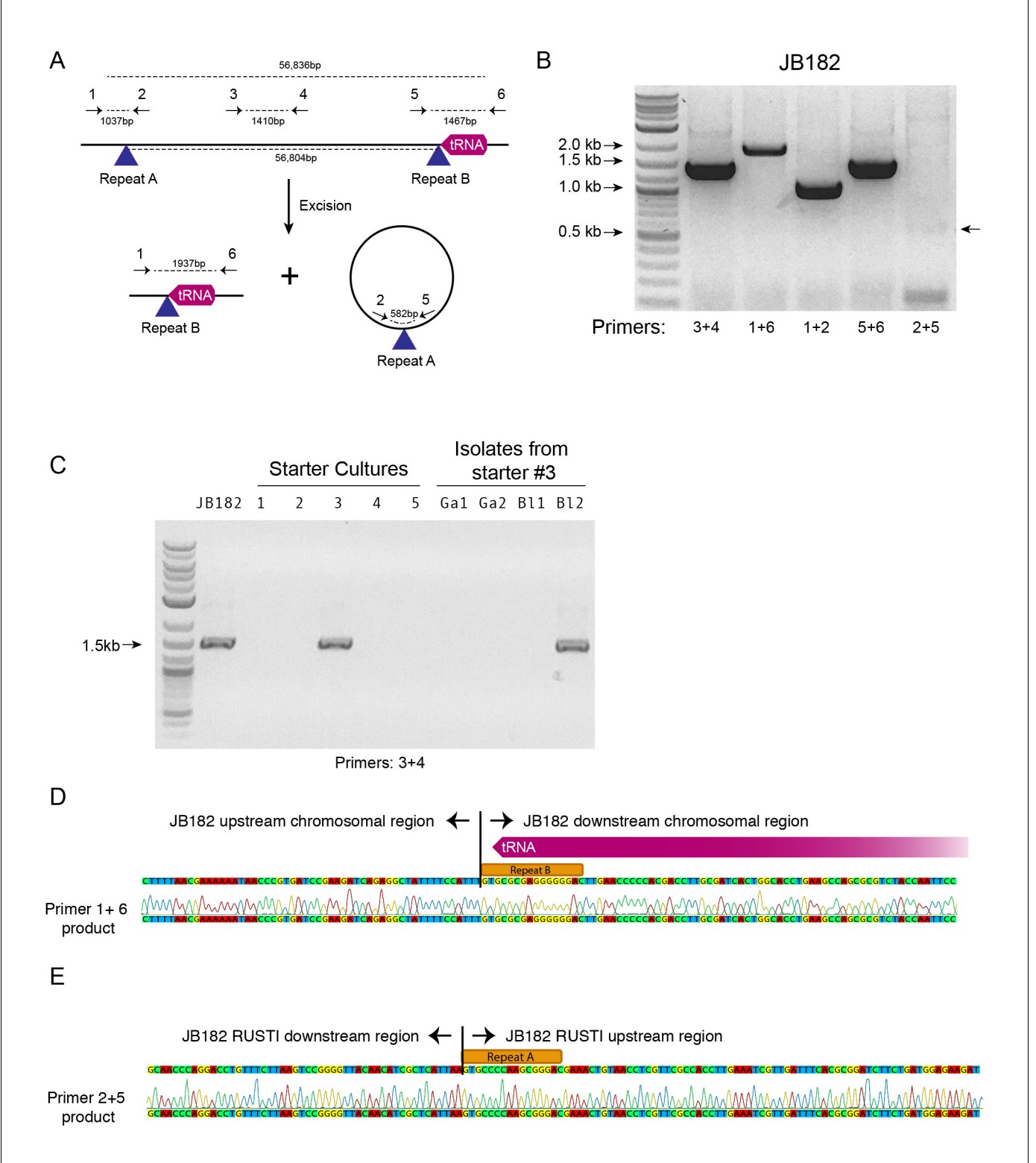

**Figure 5.** Mobility of RUSTI. (**A**) Schematic for PCR primer design - see Materials and methods for details. (**B**) PCR testing for the presence of RUSTI and for the excision of the ICE in an overnight culture of *G. arilaitensis* JB182. (**C**) DNA was extracted from five commercially available starter cultures and tested for the presence of RUSTI using PCR with primers specific for the HGT region (Materials and methods). Starter culture 3 was plated on PCAMS media, and four isolates selected based on colony morphology were also tested. The expected size for the amplicon is ~1.4 kb. Sequencing of the 16S ribosomal RNA genes for these isolates suggested that two isolates are *Glutamicibacter arilaitensis* and two are *Brevibacterium linens*. *G. arilaitensis*. *Figure 5 continued on next page*

*Figure 5 continued*

JB182 was used as a positive control. (D) The ~2000 bp band from the PCR amplification using primers 1 and 6 and (E) the ~500 bp band from amplification using primers 2 and 5 were extracted and sequenced. Alignment with the JB182 genome reveals 100% alignment with expected and the spliced chromosomal region containing the 3' repeat and the excision circle containing the 3' repeat respectively.

five common starter cultures for the presence of ActinoRUSTI by PCR, and positively identified it in one of these (*Figure 5C*). This culture is known to contain two species of Actinobacteria, *Brevibacterium linens* and *Glutamicibacter nicotinae*. To identify the RUSTI donor species, we plated the starter culture and isolated four distinct strains based on colony morphology. SSU sequencing revealed that two isolates were *Glutamicibacter* and two were *Brevibacterium*. One of the *Brevibacterium* isolates tested positive for ActinoRUSTI by PCR (*Figure 5C*). Although we have no evidence for direct transmission of ActinoRUSTI from this starter culture, and thus cannot definitively conclude that this was the source, these data are consistent with the hypothesis that HGT from a starter culture could explain some of the dissemination of ActinoRUSTI.

## Discussion

In this paper, we provide evidence of extensive horizontal gene transfer in cheese-associated bacteria. Many of the transferred regions are large multi-gene islands, and are shared by numerous species. Genes involved in nutrient acquisition, especially iron and lactate, are particularly abundant, suggesting that HGT may provide a selective advantage in the iron- and sugar-limited environment of cheese. The largest HGT we identified appears to be an active ICE and is found in a starter culture, raising the possibility that we are observing contemporary processes that may have ongoing importance. These data support previous studies that show HGT is an important factor in the evolution of microbial communities (*Wiedenbeck and Cohan, 2011*) and suggest that cheese rind communities may provide a useful model for studying this process in greater detail.

For this study, we focused on bacterial members of the cheese microbiome, but many cheeses also contain numerous fungal species. Indeed, HGT has been previously documented in cheese-associated *Penicillium* species (*Cheeseman et al., 2014*). HGT between bacteria and fungi has also been documented in other environments, although it is thought to be rare (*Keeling and Palmer, 2008*). Evaluating bacterial-fungal gene transfer in cheese could provide additional insights into the extent and importance of gene exchange in microbial communities. Further sequencing of bacterial genomes from cheese could also continue to reveal HGT.

We were able to show that ActinoRUSTI in *G. arilaitensis* JB182 is likely contained within an integrative and conjugative element, although many other mechanisms for transfer are likely at play. Indeed, the appearance of phage-related genes, transposases, and other mobile elements in many HGT groups suggests that we are observing the results of multiple methods for mobilizing DNA.

Our method for identifying HGT does not permit determination of the direction of gene flow, and indeed it seems likely that the original sources of many of these genetic elements are not present in our dataset. In some cases, we can infer a possible origin, such as the *Brevibacterium* strain in a starter culture that may be the source of RUSTI in multiple cheese species around the world. However, this does not permit identification of where this species acquired them. Further characterization of cheese-associated microbes, as well as those found in dairy farms or in cheese caves may provide a more complete picture, but the evidence that at least some of these genetic elements are found in human pathogens and ocean-dwelling bacteria suggests that genes are shared across diverse environments.

Although previous studies demonstrated that iron is limiting for *Glutamicibacter*, *Brevibacterium*, and *Corynebacterium* species growing on cheese (*Monnet et al., 2012*, *2010*), the preponderance of siderophore and other iron acquisition genes we observed being horizontally transferred suggests that the same is true across bacterial phyla. Limiting iron is a deliberate strategy on the part of mammalian hosts to block the growth of infectious microbes, and this strategy influences the composition of milk because of the presence of lactoferrin (*Ellison, 1994*). Interestingly, convergent strategies for acquiring iron are used by pathogens and by cheese-associated microbes and we observe that in some cases these disparate species appear to have shared genes through horizontal transfer. The

presence of these same genes in a microbe found in ocean habitats suggests that these genes have broad utility for the common challenge of iron limitation.

We have yet to demonstrate the functional consequences of these genes on individual species or on the community as a whole. Given that iron is limiting in cheese, and that ActinoRUSTI genes are upregulated in response to other species (*Figure 4C*), it is likely that these genes are functional and may play a role in competition. The prevalence of siderophore import, but not siderophore synthesis pathways suggests that species may cheat by scavenging the biosynthetic products of others (*Cordero et al., 2012*).

The identification of widespread sharing of genes in cheese microbial communities could have important implications. In particular, the possibility that a starter culture is the source of mobile gene elements suggests that the genomic content, rather than just specific species, must be considered when designing microbial supplements. In addition to starter cultures used for fermented foods, living microbial supplements ('probiotics') are increasingly being adopted in agriculture (*Verschuere et al., 2000*; *Chaucheyras-Durand and Durand, 2010*) and for a wide range of human health conditions (*Cuello-Garcia et al., 2015*; *Onubi et al., 2015*; *(IBS Dietetic Guideline Review Group on behalf of Gastroenterology Specialist Group of the British Dietetic Association) et al., 2016*), and even as cosmetics (*Whitlock et al., 2016*). The need to screen for clinically relevant elements such as antibiotic resistance genes is widely recognized (*Sanders et al., 2010*), but other mobile gene elements from these organisms may also enter native microbial populations with unknown consequences.

Although we and others have observes a large number of HGT events in microbial species across a diverse range of environments (*McDaniel et al., 2010*; *Smillie et al., 2011*; *Ravenhall et al., 2015*), the biotic and abiotic conditions that affect this frequency, and what effects HGT may have on the community remain unclear. A model system to study HGT in a community context is particularly important, as sequence-based characterization of complex communities has particular limitations when it comes to HGT. Further, even if complete characterization in situ were possible, many microbial communities are difficult to experimentally manipulate in vitro.

By contrast, cheese rind-associated bacteria are readily isolated and cultured, and model communities may enable identification of features of microbial communities and their environments that alter frequency and extent of HGT. The cheese rind model system provides an opportunity to observe HGT as it happens and to investigate how community composition affects the frequency of transfer and the persistence of genes. The in vitro cheese system enables experimental manipulation to investigate the role of community composition in driving HGT. Further, as many gene products may only have survival benefits in the context of community competition and cooperation, investigating the role of RUSTI and other horizontally transferred genes on microbial growth in the context of their natural community is critical. Having an experimentally tractable microbial community will allow us to test these ideas under controlled conditions in the laboratory and generate predictions about how these processes work in more complex natural systems. Many species of cheese-associated bacteria have close relatives in the soil, on skin, in the ocean, making insight from this system potentially applicable to diverse environments. In addition, horizontal acquisition of iron-uptake genes has been documented in numerous environments including the oceans and in human pathogens (*Gyles and Boerlin, 2014*; *Richards et al., 2009*), suggesting that the specific processes occurring in cheese may also be generally informative across systems. Understanding the extent of HGT in the cheese microbiome is the first step towards addressing how the movement of genes shapes, and is shaped by, a microbial community. Using cheese rinds as a model system could help elucidate the factors that influence the frequency of HGT, how it impacts competition and cooperation, and helps shape a microbiome.

## Materials and methods

### Sequencing and genome assembly

Bacterial strains JB4, 5, 7, 37, 110, 182, 196, and 197 were isolated from cheeses in a single geographic region and sequenced using a combination of Illumina short-read (100 bp, paired end) and Pacbio long-read sequencing. DNA was extracted using Genomic Tip 100/G (Qiagen, USA) or Power Soil (MoBio, USA). Illumina library preparation and sequencing were performed at Harvard University

by the Bauer Core facility. Pacbio library preparation and sequencing were performed by the University of Massachusetts Medical School Deep Sequencing Core. De novo hybrid assembly was performed using SPAdes (v3.5.0) (*Bankevich et al., 2012*). Genomes were annotated using the Integrated Microbial Genomes Expert Review (IMG/ER) annotation pipeline (*Markowitz et al., 2012*). In addition, we also sequenced eight additional rind isolates of *Brachybacterium* (strains 341.9, 738.10, 862.8, 876.9, 900.8, 908.11, 947.1, 962.10) and *Brevibacterium* (strains 341.13, 738.8, 862.7, 876.7, 900.6, 908.7, 947.7, 962.8) and three additional isolates of *Glutamicibacter* (strains BW77, 78, 80) from different cheeses in a broad geographic distribution. For these isolates, we prepared draft genomes using Illumina short-read sequencing and assembled with CLC genomics workbench.

The annotated genomes used can be found on Zenodo (*Bonham et al., 2016*)

## Phylogenetic trees

16S sequences were retrieved from the sequenced genomes and aligned using the structure-based aligner, Infernal v1.1rc4 (*Nawrocki and Eddy, 2013*), as implemented in the Ribosomal Database Project release 11 (*Cole et al., 2009*). The alignment was imported into Geneious v9 (Biomatters, LTD), and a tree was calculated using the maximum likelihood method PHYML (GTR model) (*Guindon et al., 2003*). The tree was rooted using *Thermus thermophilus*. The tree was then uploaded to Interactive Tree of Life (iTOLv3) (*Letunic and Bork, 2016*) to enable mapping of HGT data (connections and group abundance profiles).

## RNAseq

Four replicate transcriptomes from two treatments were sequenced: (1) *G. arilaitensis* alone and (2) *G. arilaitensis* + *Penicillium*. We used a strain of *Penicillium solitum* that was isolated from a natural rind cheese and was used for experiments in *Wolfe et al. (2014)*. For each experimental unit, approximately 80,000 CFUs of *Glutimicibacter arilaitensis* were spread across the surface of a 100 mm Petri dish containing 20 mL of cheese curd agar (10% freeze-dried fresh cheese, 3% NaCl, 1.7% agar, 0.5% xanthan gum) (*Wolfe et al., 2014*). For the + *Penicillium* treatment, approximately 100,000 CFUs were co-inoculated onto the plates with the *Glutimicibacter*. Plates were incubated in a bin with moist paper towel (>90% relative humidity) at 24°C for 5 days.

Rind biofilms were then harvested by scraping the cheese curd surface and stored in RNAProtect Reagent (Qiagen) to stabilize mRNA frozen at −80°C. RNA was extracted using a standard phenol-chloroform protocol used for many different fungal and bacterial species, which has been adopted from transcriptomics work in gut microbiomes (see [*David et al., 2014*]). This protocol uses a standard bead-beating step in a lysis buffer to release cell contents from biofilms stored in RNAProtect. DNA was removed from the samples using a TURBO DNA-free kit (Life Technologies), and 5S tRNA and large rRNA was depleted using MEGAClear (Life Technologies) and RiboZero (Illumina) kits, respectively. To remove both fungal and bacterial large rRNA, we used an equal mixture of RiboZero Yeast and Bacteria rRNA Removal Solution. To confirm that the samples were free of DNA contaminants, a PCR of the 16S rRNA gene was performed with standard primers (27 f and 1492 r). Overall quantity and quality of the RNA preps were confirmed by Nanodrop and Agilent 2100 Bioanalyzer using the RNA 6000 Nano kit.

RNA-seq libraries were constructed from purified mRNA using the NEBNext Ultra RNA Library Prep Kit for Illumina (New England Biolabs) where each library received a unique six base pair barcode for demultiplexing after the sequencing run. Each library was run on an Agilent 2100 Bioanalyzer High Sensitivity DNA chip to confirm that primer dimers and adapter dimers were not present in the sample and to determine the size of the library. Final libraries were standardized to 10 nM each after quantification with a Qubit dsDNA HS Assay Kit (Life Technologies) and then pooled in equal amounts to get similar sequencing coverage across all libraries. The pooled library samples were sequenced using paired-end 100 bp reads on an Illumina HiSeq Rapid Run by the Harvard Bauer Core Sequencing Core Facility.

To quantify gene expression and determine whether genes within RUSTI were differentially expressed when grown with the competitor *Penicillium*, we used Rockhopper (*McClure et al., 2013*). Only forward reads were used for this analysis. The assembled and annotated *Glutamicibacter arilaitensis* strain JB182 (described above) genome were used as a reference genome for

mapping. We considered genes that had a greater than fourfold difference in expression when grown with *Penicillium* or *Staphylococcus* and were significantly different (based on Rockhopper's *q*-values, which control for false discovery rate using the Benjamini-Hochberg procedure) to be differentially expressed genes.

## PCR

PCR reactions were performed using Q5 Hot Start Mastermix (New England Biolabs). Where JB182 RUSTI is integrated in the chromosome, PCR using primer 1 (CAACTGTGCCACGCAATTCA) and primer 2 (CGGCTACTTCTCGGATGGTC) are expected to produce a 1037 bp product that includes the 5' ICE repeat. Primer 3 (CGCAATCGTGGTGTATCTGC) and primer 4 (GACGGGATCAGGAAC-GACG) should produce a 1410 bp product, while primer 5 (GCCGCATCTACCTCGATGAA) and primer 6 (CCAAATCGCGACGCATTGAT) are expected to form a 1467 bp product. Primers 1 and 6 are separated by approximately 59 kb when RUSTI is present and are not expected to form a PCR product, but should form a 1937 bp product if RUSTI is excised. Primers 2 and 5 should not form a PCR product when RUSTI is integrated, but would form a 500 bp product if the excision circle were present.

## Additional software

Annotated genomes were compared using blastn from BLAST+ (v2.3.0) (*Camacho et al., 2009*). Protein coding genes were considered to be potential HGT if their sequence was at least 99% identical for at least 500 nucleotides. Neighboring candidate HGT were identified as part of the same island if they were separated by no more than 5000 nucleotides. Scripts to import and store genome information and blast results and to analyze results are available on github (*Bonham, 2016*; https://github.com/kescobo/kvasir). A copy is archived at https://github.com/elifesciences-publications/kvasir.

Genomic average nucleotide identity (ANI) was calculated using the 'ani.rb' script from the enveomics collection (commit 'e8faed01ff848222afb5820595cccc4e50c89992') with default settings (*Rodriguez-R and Konstantinidis, 2016*).

## Metagenomes

Shotgun metagenomic data from (*Wolfe et al., 2014*) and (*Kastman et al., 2016*) were assembled with CLC Genomic Workbench 8.0. Representative sequences for ActinoRUSTI or ProteoRUSTI were compared with assembled metagenomes by BLAST. Hits with >97% similarity were considered to be positive hits for target regions.

## Accession numbers

Newly sequenced genomes were registered with NCBI with the bioproject ID PRJNA387187. Biosample accession numbers for individual genomes are shown in *Supplementary file 1a*, and are as follows:

*Brevibacterium linens* 341_13: SAMN07141149, *Brevibacterium linens* 738_8: SAMN07141150, B*revibacterium linens* 862_7: SAMN07141151, *Brevibacterium linens* 876_7: SAMN07141152, *Brevibacterium linens* 900_6: SAMN07141153, *Brevibacterium linens* 908_7: SAMN07141154, *Brevibacterium linens* 947_7: SAMN07141155, *Brevibacterium linens* 962_8: SAMN07141156, *Brachybacterium alimentarium* 341_9: SAMN07141157, *Brachybacterium alimentarium* 738_10: SAMN07141158, *Brachybacterium alimentarium* 862_8: SAMN07141159, *Brachybacterium alimentarium* 876_9: SAMN07141160, *Brachybacterium alimentarium* 900_8: SAMN07141161, *Brachybacterium alimentarium* 908_11: SAMN07141162, *Brachybacterium alimentarium* 947_11: SAMN07141163, *Brachybacterium alimentarium* 962_10: SAMN07141164, *Glutamicibacter* sp. BW77: SAMN07141165, *Glutamicibacter* sp. BW78: SAMN07141166, *Glutamicibacter* sp. BW80: SAMN07141167, *Microbacterium* sp. JB110: SAMN07141168, *Halomonas* sp JB37: SAMN07141169, *Brevibacterium linens* JB5: SAMN07141170, *Psychrobacter* sp. JB193: SAMN07141171, *Brachybacterium* sp. JB7: SAMN07141172, *Sphingobacterium* sp. JB170: SAMN07141173, *Vibrio casei* JB196: SAMN07141174, *Arthrobacter* sp. JB182: SAMN07141175, *Corynebacterium* sp. JB4: SAMN07141176, *Pseudoalteromonas* sp. JB197: SAMN07141177.

## Acknowledgements

The authors would like to thank Rajashree Mishra for assisting with initial stages of project development, Dana Boyd for assistance with ICE identification, Brian Tsu for assistance with the Transporter Classification Database, and members of the Dutton lab for helpful conversations.

## Additional information

### Funding

| Funder | Grant reference number | Author |
|---|---|---|
| National Institutes of Health | P50 GM068763 | Kevin S Bonham<br>Benjamin E Wolfe<br>Rachel J Dutton |

The funders had no role in study design, data collection and interpretation, or the decision to submit the work for publication.

### Author contributions

KSB, RJD, Conceptualization, Investigation, Writing—original draft, Writing—review and editing; BEW, Conceptualization, Investigation, Writing—review and editing

### Author ORCIDs

Kevin S Bonham, http://orcid.org/0000-0003-3200-7533
Rachel J Dutton, http://orcid.org/0000-0002-2944-2182

## Additional files

### Supplementary files

• Supplementary file 1. Genome information. (a) Genome statistics for newly sequenced genomes, determined by IMG/ER. Gene IDs refer to IMG bioproject or RefSeq Accession. Genomes from Almeida et. al. do not yet have accession numbers. (b) Pairwise species comparison summary total. Protein coding sequences (column 'Shared CDS') and nucleotides (in base-pairs - column 'Shared nt') determined to be horizontally transferred for every pair of species that were compared by the HGT detection pipeline. Also shows calculated ANI and 16S similarity in % (column ssu - see Materials and methods for method of determining 16S similarity). Species pairs that have ANI > 0.89 were not compared and are not shown. (c) HGT identification parameters. Different parameters for minimum length of gene match for HGT, maximum % ANI identity for related species, and maximum distance between genes in an island were compared. Number of positive HGT hits identified when varying the minimum protein coding gene length. Number of HGT groups constructed when varying the maximum separation between hits that are classified as belonging to the same group. Number of nucleotides or number of protein coding sequences in HGT regions by 16S similarity. Note - There are no results below 500 as 500 bp is the minimum length for protein coding sequences in this analysis. (d) Full group annotations. All protein coding sequences identified as HGT, sorted by group # (ranked by total nucleotide content), species and genome location within species. Certain functional annotations are identified by color (e.g. orange for iron) based on text in annotation. Locus tags and contig IDs beginning with lower case letters were assigned by kvasir, and do not correspond to any published database. (e) Group summary statistics. Summary statistics for each HGT group. (f) Highly conserved genes in *Brevibacterium* species. Protein coding sequences from Group 29, as well as selected highly conserved genes from *Brevibacterium antiquum* CNRZ918 were compared with other *Brevibacterium* strains by BLAST. *B. linens* 947.7 has substantially lower nucleotide identity for the four genes found in Group 29 than other *B. linens* strains, despite similar nt distance for other highly conserved genes. This suggests that Group 29 is a true example of HGT between CNRZ918 and other *B. linens* strains, rather than a false positive. (g) RUSTI gene expression during competition. Gene expression data from RNA seq analysis for genes in JB182 RUSTI. Related to *Figure 3B* (h) TCBD hits for transporters in RUSTI. Representative CDS of Actino- and ProteoRUSTI from *G. arilaitensis* JB182 and *V. casei* JB196, respectively, were compared with the Transporter Classification

Database (TCDB). (i) RefSeq BLAST Actino- and ProteoRUSTI from *G. arilaitensis* JB182 and *V. casei* JB196, respectively, as well as the consensus sequence for StaphRUSTI (see *Figures 3* and *4*) were compared with the NCBI RefSeq database using BLAST.

## Major datasets

The following datasets were generated:

| Author(s) | Year | Dataset title | Dataset URL | Database, license, and accessibility information |
|---|---|---|---|---|
| Bonham KS, Wolfe BE, Dutton RJ | 2016 | Datasets associated with Bonham et al. | https://doi.org/10.5281/zenodo.163212 | Publicly available at Zenodo (https://zenodo.org) |
| Bonham KS, Wolfe BE, Dutton RJ | 2017 | Datasets associated with Bonham et al. | http://www.ncbi.nlm.nih.gov/bioproject/387187 | Publicly available at the NCBI BioProject (accession no: PRJNA387187) |

The following previously published datasets were used:

| Author(s) | Year | Dataset title | Dataset URL | Database, license, and accessibility information |
|---|---|---|---|---|
| Wolfe BE, Dutton RJ | 2014 | shotgun metagenomic data from cheese rinds used in Figure 4 | http://metagenomics.anl.gov/mgmain.html?mgpage=overview&metagenome=mgm4524487.3 | Publicly available at MG-RAST (accession no: 4524487.3) |
| Wolfe BE, Dutton RJ | 2014 | shotgun metagenomic data from cheese rinds used in Figure 4 | http://metagenomics.anl.gov/mgmain.html?mgpage=overview&metagenome=mgm4524500.3 | Publicly available at MG-RAST (accession no: 4524500.3) |
| Wolfe BE, Dutton RJ | 2014 | shotgun metagenomic data from cheese rinds used in Figure 4 | http://metagenomics.anl.gov/mgmain.html?mgpage=overview&metagenome=mgm4524498.3 | Publicly available at MG-RAST (accession no: 4524498.3) |
| Wolfe BE, Dutton RJ | 2014 | shotgun metagenomic data from cheese rinds used in Figure 4 | http://metagenomics.anl.gov/mgmain.html?mgpage=overview&metagenome=mgm4524496.3 | Publicly available at MG-RAST (accession no: 4524496.3) |
| Wolfe BE, Dutton RJ | 2014 | shotgun metagenomic data from cheese rinds used in Figure 4 | http://metagenomics.anl.gov/mgmain.html?mgpage=overview&metagenome=mgm4524502.3 | Publicly available at MG-RAST (accession no: 4524502.3) |
| Wolfe BE, Dutton RJ | 2014 | shotgun metagenomic data from cheese rinds used in Figure 4 | http://metagenomics.anl.gov/mgmain.html?mgpage=overview&metagenome=mgm4524495.3 | Publicly available at MG-RAST (accession no: 4524495.3) |
| Wolfe BE, Dutton RJ | 2014 | shotgun metagenomic data from cheese rinds used in Figure 4 | http://metagenomics.anl.gov/mgmain.html?mgpage=overview&metagenome=mgm4524488.3 | Publicly available at MG-RAST (accession no: 4524488.3) |
| Wolfe BE, Dutton RJ | 2014 | shotgun metagenomic data from cheese rinds used in Figure 4 | http://metagenomics.anl.gov/mgmain.html?mgpage=overview&metagenome=mgm4524490.3 | Publicly available at MG-RAST (accession no: 4524490.3) |
| Wolfe BE, Dutton RJ | 2014 | shotgun metagenomic data from | http://metagenomics.anl. | Publicly available at |

| | | | | |
|---|---|---|---|---|
| | | | cheese rinds used in Figure 4 | gov/mgmain.html?<br>mgpage=overview&me-<br>tagenome=<br>mgm4524499.3 | MG-RAST (accession<br>no: 4524499.3) |
| Wolfe BE, Dutton RJ | 2014 | shotgun metagenomic data from<br>cheese rinds used in Figure 4 | http://metagenomics.anl.<br>gov/mgmain.html?<br>mgpage=overview&me-<br>tagenome=<br>mgm4524497.3 | Publicly available at<br>MG-RAST (accession<br>no: 4524497.3) |
| Wolfe BE, Dutton RJ | 2014 | shotgun metagenomic data from<br>cheese rinds used in Figure 4 | http://metagenomics.anl.<br>gov/mgmain.html?<br>mgpage=overview&me-<br>tagenome=<br>mgm4524491.3 | Publicly available at<br>MG-RAST (accession<br>no: 4524491.3) |
| Wolfe BE, Dutton RJ | 2014 | shotgun metagenomic data from<br>cheese rinds used in Figure 4 | http://metagenomics.anl.<br>gov/mgmain.html?<br>mgpage=overview&me-<br>tagenome=<br>mgm4524493.3 | Publicly available at<br>MG-RAST (accession<br>no: 4524493.3) |
| Wolfe BE, Dutton RJ | 2014 | shotgun metagenomic data from<br>cheese rinds used in Figure 4 | http://metagenomics.anl.<br>gov/mgmain.html?<br>mgpage=overview&me-<br>tagenome=<br>mgm4524501.3 | Publicly available at<br>MG-RAST (accession<br>no: 4524501.3) |
| Wolfe BE, Dutton RJ | 2014 | shotgun metagenomic data from<br>cheese rinds used in Figure 4 | http://metagenomics.anl.<br>gov/mgmain.html?<br>mgpage=overview&me-<br>tagenome=<br>mgm4524482.3 | Publicly available at<br>MG-RAST (accession<br>no: 4524482.3) |
| Wolfe BE, Dutton RJ | 2014 | shotgun metagenomic data from<br>cheese rinds used in Figure 4 | http://metagenomics.anl.<br>gov/mgmain.html?<br>mgpage=overview&me-<br>tagenome=<br>mgm4524489.3 | Publicly available at<br>MG-RAST (accession<br>no: 4524489.3) |
| Wolfe BE, Dutton RJ | 2014 | shotgun metagenomic data from<br>cheese rinds used in Figure 4 | http://metagenomics.anl.<br>gov/mgmain.html?<br>mgpage=overview&me-<br>tagenome=<br>mgm4524483.3 | Publicly available at<br>MG-RAST (accession<br>no: 4524483.3) |
| Wolfe BE, Dutton RJ | 2014 | shotgun metagenomic data from<br>cheese rinds used in Figure 4 | http://metagenomics.anl.<br>gov/mgmain.html?<br>mgpage=overview&me-<br>tagenome=<br>mgm4524505.3 | Publicly available at<br>MG-RAST (accession<br>no: 4524505.3) |
| Wolfe BE, Dutton RJ | 2014 | shotgun metagenomic data from<br>cheese rinds used in Figure 4 | http://metagenomics.anl.<br>gov/mgmain.html?<br>mgpage=overview&me-<br>tagenome=<br>mgm4524494.3 | Publicly available at<br>MG-RAST (accession<br>no: 4524494.3) |
| Wolfe BE, Dutton<br>RJ | 2014 | shotgun metagenomic data from<br>cheese rinds used in Figure 4 | http://metagenomics.anl.<br>gov/mgmain.html?<br>mgpage=overview&me-<br>tagenome=<br>mgm4524486.3 | Publicly available at<br>MG-RAST (accession<br>no: 4524486.3) |
| Wolfe BE, Dutton RJ | 2014 | shotgun metagenomic data from<br>cheese rinds used in Figure 4 | http://metagenomics.anl.<br>gov/mgmain.html?<br>mgpage=overview&me-<br>tagenome=<br>mgm4524504.3 | Publicly available at<br>MG-RAST (accession<br>no: 4524504.3) |
| Wolfe BE, Dutton RJ | 2014 | shotgun metagenomic data from<br>cheese rinds used in Figure 4 | http://metagenomics.anl.<br>gov/mgmain.html?<br>mgpage=overview&me-<br>tagenome=<br>mgm4524485.3 | Publicly available at<br>MG-RAST (accession<br>no: 4524485.3) |
| Wolfe BE, Dutton RJ | 2014 | shotgun metagenomic data from<br>cheese rinds used in Figure 4 | http://metagenomics.anl.<br>gov/mgmain.html? | Publicly available at<br>MG-RAST (accession |

mgpage=overview&me-  no: 4524484.3)
tagenome=
mgm4524484.3

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
