## [Decision Letter]

Thank you for submitting your article "Extensive Horizontal Gene Transfer in Cheese-Associated Bacteria" for consideration by *eLife*. Your article has been favorably evaluated by Wendy Garrett (Senior Editor) and three reviewers, one of whom, Kim Handley (Reviewer #1), served as Guest Reviewing Editor. The following individuals involved in review of your submission have agreed to reveal their identity: Danilo Ercolini (Reviewer #2) and Jennifer Martiny (Reviewer #3).

The reviewers have discussed the reviews with one another and the Reviewing Editor has drafted this decision to help you prepare a revised submission.

Summary:

The reviewers agree that this is an interesting and timely study on bacterial HGT using cheese rinds as a model system. The reviewers also raised some important points that should be addressed in a revised version as summarized below.

Essential revisions:

Please provide further evidence to support the conclusions that the gene groups identified do in fact represent HGT as commented on by reviewers 1 and 3. In order to keep the conclusion that HGT is prevalent in cheese rind communities it is also necessary to provide a frame of reference or control group, for example how does this compare in other systems? Reviewer 2 also has made some reasonable requests to add some caveats or temper (tone down) statements in places, e.g. regarding the starting culture as a source of HGT, although we do agree this is an interesting and convincing finding.

*Reviewer #1:*

The study uses cheese rinds as model environments in which to study microbial HGT. It identifies HGT among a large number of geographically diverse cheese rind microorganisms, and shows that many acquired islands contain genes important for Fe acquisition in an Fe limited environment. The authors demonstrate a prevalent HGT island is actively mobile, and is also present in a commonly used starter culture indicating a possible mechanism for global distribution. This is a well thought out and thorough study that advances our understanding of HGT in microbial communities.

A major conclusion of the study is that HGT is high in cheese rind communities. However, there is no point of reference on which to base this conclusion. High compared to what?

The authors identified mobile elements and insertion sites indicative of genomic islands for one HGT group (RUSTI). It would be useful to know (and strengthen the study) what other HGT groups identified are associated with mobile elements, and what proportion of the total groups. Also Given bacterial genomes can contain a number of putative genomic islands, what is the proportion identified HGT groups to genomic islands not horizontally transferred among the organisms compared?

*Reviewer #2:*

The topic is timely, the results are very interesting and adequately discussed.

HGT is "the" fascinating source of microbial diversity and here we can see how such diversity can be potentially spread in the complex communities of cheese rinds and how it could play an important role in adaptation to the environment.

I have the following considerations and a few editing suggestions I would expect the authors will consider in revising their manuscript.

“During the aging of cheese, bacteria and fungi form a multi-species biofilm called the rind [Button and Dutton, 2012]”. Such a statement must be changed. This does not apply to just any cheese. So please amend indicating in what types of cheese this happens.

Subsection “Iron Acquisition HGT”, sixth paragraph. This is unlikely to play a role in adaptation, at least in the case of cheese. In addition, the authors do not discuss the Staph HTG. I would just leave out this part. Overall it stands as an awkward deviation in a nice story.

Subsection “A potential mechanism of transfer and source of the Actinobacterial RUSTI”, last paragraph. Although this is a convincing initial indication, it should be proven in a remarkably higher number of cases. I recommend the authors to down-tone the statements in this paragraph to make the conclusions more careful and scaled to the only experiment and case reported.

Discussion, seventh paragraph. This is far too alarming for the readership. HTG may just happily take place naturally in any ecosystem. I suggest rethinking this bit of Discussion.

Discussion, ninth paragraph. Honestly I do not believe in in vitro models. I know the authors already have experience with this. My opinion is that what happens in reconstituted communities with no cheese around can be very different from what could take place in a real system. I would expect the authors to at least recognize the limitations of in vitro studies of food communities in this part of the Discussion.

*Reviewer #3:*

This article investigates evidence for horizontal gene transfer among bacterial genomes associated with cheese. The manuscript is well-written, the figures are clear, and the conclusions are well-supported. It is not surprising that HGT is prevalent among these bacteria, but it is useful to consider what types of genes are involved to develop cheese as a model system.

I don't have major substantive comments, but did wonder whether the authors had considered doing any analysis of the genomic location of the HGT groups, at least for the large groups. If similar groups are found in very different genomic locations, even among closely related strains, then this would provide evidence for HGT. For instance, this type of analysis would be useful for group 27 and its presence among the *B. linens* strains (subsection “Functions encoded in HGT regions”, sixth paragraph). Our group has been trying to consider these patterns for HGT in bacteriophage (http://www.sciencedirect.com/science/article/pii/S0042682216302756), and perhaps some consideration of this might help to support or refute the alternatives proposed. However, this is not essential for publication.

---

## [Author Response]

*Essential revisions:*

*Please provide further evidence to support the conclusions that the gene groups identified do in fact represent HGT as commented on by reviewers 1 and 3.*

In order to support the conclusion that the groups identified in the manuscript represent HGT, reviewer 1 requests that we report the extent to which our HGT groups contain genes associated with mobile elements. In our analysis of gene functions encoded by the HGT regions, we found that gene functions associated with mobile genetic elements are the most abundant function identified:

“the most abundant gene functions are transposases, conjugal transfer genes, phage-related proteins and other mobile elements (631/4733 or 14% of all protein coding sequences). Nearly a third (86/264 or 32.6%) of all HGT groups contain mobile elements. These genetic elements are likely involved in either directly or indirectly in the mobilization and transfer of DNA.”

We have now added an additional column in [Supplementary-material SD1-data] to indicate which of the HGT groups identified in our study contain genes associated with mobile elements and an additional column to [Supplementary-material SD1-data] which indicates the number of HGT groups found in each genome that we sequenced. Unfortunately, since we are looking only at recent transfer events (by filtering on genes that are >99% identical), and since our method would not identify islands that are horizontally acquired from an organism that we have not sequenced, we are certainly undercounting HGT in these genomes, and there are likely many more genomic islands present. Reviewer 3 suggested analyzing the genomic location of the HGT groups, with the idea that different locations in different genomes would support HGT. While we think this could be interesting to analyze, the presence of an HGT group at the same genomic location in multiple genomes does not rule out HGT. Certain mobile elements, such as integrative and conjugative elements, are known to integrate at the same genomic site due to conserved sequences (such as tRNAs). Also, HGT that occurs via transformation and homologous recombination should not lead to changes in the genomic location of groups.

*In order to keep the conclusion that HGT is prevalent in cheese rind communities it is also necessary to provide a frame of reference or control group, for example how does this compare in other systems?*

In this study, we identified several hundred potential horizontally transferred regions mapping to two or more genomes amongst the 165 genomes we analyzed. In some cases, these regions were found in numerous genomes (33 genomes in two cases). For this reason, we referred to HGT as being “extensive” in this dataset. However, as the reviewers point out, it is extremely helpful to put this into context of other environments.

Indeed, previous studies indicate that HGT can occur very frequently in natural ecosystems such as the ocean, the human microbiome, and in particular within biofilms (and we have added additional references to the text to highlight this point). As a result, it was not our intention to claim that there is more HGT in cheese than in other environments, merely that we find many examples of HGT in cheese-associated species. We thank the reviewers for bringing this point to our attention, and we have included the following into the paper, which we hope both adds context to our work by providing a frame of reference, and prevents any confusion:

“Advances in genome sequencing have begun to provide a glimpse into HGT within environmentally, medically and economically important microbiomes [McDaniel et al., 2010; Andam, Carver and Berthrong, 2015]. […] Other natural habitats, such as soil (Coombs and Barkay 2004; Heuer and Smalla 2012) and aquatic environments (McDaniel et al. 2010; Frischer et al. 1994) also show evidence of extensive HGT.”

*Reviewer 2 also has made some reasonable requests to add some caveats or temper (tone down) statements in places, e.g. regarding the starting culture as a source of HGT, although we do agree this is an interesting and convincing finding.*

We have added additional language to make it clear that these findings are not definitive. See also specific responses to reviewer 2 below.

*Reviewer #1:*

*[…] A major conclusion of the study is that HGT is high in cheese rind communities. However, there is no point of reference on which to base this conclusion. High compared to what?*

Please see our second response to the Editor regarding this question.

*The authors identified mobile elements and insertion sites indicative of genomic islands for one HGT group (RUSTI). It would be useful to know (and strengthen the study) what other HGT groups identified are associated with mobile elements, and what proportion of the total groups. Also Given bacterial genomes can contain a number of putative genomic islands, what is the proportion identified HGT groups to genomic islands not horizontally transferred among the organisms compared?*

We agree that this an important piece of information. Please see our first response to the Editor.

*Reviewer #2:*

*[…] I have the following considerations and a few editing suggestions I would expect the authors will consider in revising their manuscript.*

*“During the aging of cheese, bacteria and fungi form a multi-species biofilm called the rind [Button and Dutton, 2012]”. Such a statement must be changed. This does not apply to just any cheese. So please amend indicating in what types of cheese this happens.*

Thank you for pointing out that this could use some clarification. We have added an additional statement to this sentence to clarify that the formation of multi-species biofilms on the surface of cheese only happens in traditional styles of rinded cheeses: “During the aging of traditional styles of cheese in caves or aging rooms, bacteria and fungi form a multi-species biofilm called the rind”.

*Subsection “Iron Acquisition HGT”, sixth paragraph. This is unlikely to play a role in adaptation, at least in the case of cheese. In addition, the authors do not discuss the Staph HTG. I would just leave out this part. Overall it stands as an awkward deviation in a nice story.*

Thank you for this comment – we’ve added additional language in an attempt to clarify this point (see below). We agree that genes such as FnBP are not likely to be adaptive in cheese, but their association with this HGT island suggests that it may have been acquired from a pathogenic strain of *Staphylococcus*. Since we also saw an iron-acquisition HGT island in γ-Proteobacteria that is identical to genes in pathogenic *E. coli*, our intent was to highlight the convergence of iron acquisition in adaptation to cheese as well as in pathogenesis.

“It is unlikely that these genes could provide a selective advantage to cheese-associated Staphylococci, but Group 31 also contains genes for iron acquisition. […] Since iron acquisition genes may be adaptive both on cheese and in pathogenesis, it is possible that this region was acquired from an animal pathogen, and the virulence genes have been preserved due to their association with these genes.”

*Subsection “A potential mechanism of transfer and source of the Actinobacterial RUSTI”, last paragraph. Although this is a convincing initial indication, it should be proven in a remarkably higher number of cases. I recommend the authors to down-tone the statements in this paragraph to make the conclusions more careful and scaled to the only experiment and case reported.*

We agree – it was not our intention to suggest that our experiments are a definitive demonstration of the source. We’ve added an additional statement to clarify this:

“Though we have no evidence for direct transmission of ActinoRUSTI from this starter culture, and thus cannot definitively conclude that this was the source, these data are consistent with the hypothesis that HGT from a starter culture could explain some of the dissemination of ActinoRUSTI.”

*Discussion, seventh paragraph. This is far too alarming for the readership. HTG may just happily take place naturally in any ecosystem. I suggest rethinking this bit of Discussion.*

We agree that this section may be somewhat alarming, we feel that it is nevertheless important to highlight this implication. HGT is known to occur in many environments, and starter cultures and probiotics are not typically screened for mobile elements other than antibiotic resistance plasmids.

*Discussion, ninth paragraph. Honestly I do not believe in in vitro models. I know the authors already have experience with this. My opinion is that what happens in reconstituted communities with no cheese around can be very different from what could take place in a real system. I would expect the authors to at least recognize the limitations of in vitro studies of food communities in this part of the Discussion.*

As with any model system, there are certainly limitations. Model systems are widely used in biology to address questions that are difficult to address in situ, our intent here is to indicate the ways in which our cheese model system has such utility in the study of HGT. However, we agree that model systems often do not fully encompass the complexity seen in natural systems, but we believe they provide an opportunity to generate predictions about the mechanisms and impacts of processes in more complex systems. To clarify this, we have added an additional statement to this paragraph:

“Having an experimentally-tractable microbial community will allow us to test these ideas under controlled conditions in the laboratory and generate predictions about how these processes work in more complex natural systems.”

*Reviewer #3:*

*[…] I don't have major substantive comments, but did wonder whether the authors had considered doing any analysis of the genomic location of the HGT groups, at least for the large groups. If similar groups are found in very different genomic locations, even among closely related strains, then this would provide evidence for HGT. For instance, this type of analysis would be useful for group 27 and its presence among the B. linens strains (subsection “Functions encoded in HGT regions”, sixth paragraph). Our group has been trying to consider these patterns for HGT in bacteriophage (http://www.sciencedirect.com/science/article/pii/S0042682216302756), and perhaps some consideration of this might help to support or refute the alternatives proposed. However, this is not essential for publication.*

Thank you for this suggestion and we agree that this could be an interesting way to look at the data, especially for Group 27 (now reassigned to be Group 29), which appears to contain genes related to core, conserved functions. It appears that Group 29 is in the same genomic location across the genomes. One potential explanation for this could be recombination instead of mobile element-based HGT. Another potential explanation is that mobile elements, such as integrative and conjugative elements, often integrate at conserved sites in the bacterial chromosome, such as at specific tRNAs. Indeed, immediately downstream of Group 29 (less than 200bp) is a tRNA-Leu ORF.